# A distributed brain response predicting the facial expression of acute nociceptive pain

Marie-Eve Picard[1,2]*, Miriam Kunz[3], Jen-I Chen[1,2], Michel-Pierre Coll[4], Etienne Vachon-Presseau[5,6,7], Tor D Wager[8], Pierre Rainville[2,9]

[1]Department of Psychology, Université de Montréal, Montreal, Canada; [2]Centre de recherche de l'institut universitaire de gériatrie de Montréal, Montreal, Canada; [3]Department of medical psychology and sociology, Medical faculty, University of Augsburg, Augsburg, Germany; [4]School of Psychology, Université Laval, Quebec, Canada; [5]Faculty of Dentistry, McGill University, Montreal, Canada; [6]Department of Anesthesia, McGill University, Montreal, Canada; [7]Alan Edwards Centre for Research on Pain, McGill University, Montreal, Canada; [8]Department of Psychological and Brain Sciences, Dartmouth College, Hanover, United States; [9]Stomatology Department, Faculté de médecine dentaire, Université de Montréal, Montreal, Canada

*For correspondence:
marie-eve.picard.2@umontreal.
ca

Competing interest: The authors declare that no competing interests exist.

## eLife assessment

Picard et al. propose a Facial Expression Pain Signature (FEPS) derived from functional magnetic resonance imaging (fMRI) data to predict facial expressions associated with painful heat stimulation. This **important** work advances our understanding of the brain mechanisms associated with facial expressions of pain. It provides **solid** evidence that facial expressions of pain contain information that is complementary to other pain-related brain processes. The work will be of broad interest to researchers from varied fields ranging from neurosciences to psychology and affective sciences.

**Abstract** Pain is a private experience observable through various verbal and non-verbal behavioural manifestations, each of which may relate to different pain-related functions. Despite the importance of understanding the cerebral mechanisms underlying those manifestations, there is currently limited knowledge of the neural correlates of the facial expression of pain. In this functional magnetic resonance imaging (fMRI) study, noxious heat stimulation was applied in healthy volunteers and we tested if previously published brain signatures of pain were sensitive to pain expression. We then applied a multivariate pattern analysis to the fMRI data to predict the facial expression of pain. Results revealed the inability of previously developed pain neurosignatures to predict the facial expression of pain. We thus propose a facial expression of pain signature (FEPS) conveying distinctive information about the brain response to nociceptive stimulations with minimal or no overlap with other pain-relevant brain signatures associated with nociception, pain ratings, thermal pain aversiveness, or pain valuation. The FEPS may provide a distinctive functional characterization of the distributed cerebral response to nociceptive pain associated with the socio-communicative role of non-verbal pain expression. This underscores the complexity of pain phenomenology by reinforcing the view that neurosignatures conceived as biomarkers must be interpreted in relation to the specific pain manifestation(s) predicted and their underlying function(s). Future studies should explore other pain-relevant manifestations and assess the specificity of the FEPS against simulated pain expressions and other types of aversive or emotional states.

## Introduction

Theories of pain communication highlight the diversity in pain manifestations, which occur through multiple channels: verbal reports, vocal complaints, changes of postures, and facial expressions. From an evolutionary perspective, several manifestations of pain appear to be preserved across vertebrate phyla and reflect various functional roles to preserve the integrity of the organism (*Sneddon, 2019*). Withdrawal behaviour allows the individual to move away from the noxious source, while facial expressions in social species convey information about the presence of a potential threat and an appeal for assistance (*Hadjistavropoulos et al., 2011*). Functionally distinct manifestations imply at least partly segregated neurophysiological processing (*Sliwa et al., 2022*). Previous fMRI studies investigating the neural correlates of acute pain have suggested that spontaneous or induced fluctuations in pain facial expression partly reflect changes in activity within the cortical targets of the spino-thalamo-cortical pathways. *Kunz et al., 2011* found a positive association between facial responses to pain and the activity in the posterior insula, the primary somatosensory area, and the anterior cingulate cortex. These fluctuations are, however, independent of changes in stimulus intensity and are inversely related to activity in prefrontal regions (*Kunz et al., 2011*; *Vachon-Presseau et al., 2016*; *Kunz et al., 2020*). This suggests that pain facial expression may reflect the integration of activity across distributed brain networks processing ascending nociceptive signals, determining action policy, and gating efferent facial motor outputs (see *Kunz et al., 2011* and *Kunz et al., 2020* for further discussion).

The interest in developing pain neuro markers has led researchers to use multivariate pattern analysis to investigate the distributed brain mechanisms underlying the experience of pain evoked by acute nociceptive stimuli. However, fMRI studies have revealed not one but several brain signatures of acute experimental pain that may reflect the diversity and complexity of pain-related function. The neurological pain signature (NPS; *Wager et al., 2013*) was developed to predict changes in pain induced by variations in stimulus intensity and captured by subjective reports, reflecting primarily the cerebral contributions to acute nociceptive pain (*Krishnan et al., 2016*; *Wager et al., 2013*). To account for spontaneous fluctuations in the perception of pain intensity, the stimulus-independent intensity of pain signature (SIIPS-1) was trained on noxious thermal trials after statistically removing the effects of the stimulus intensity and the NPS response (*Woo et al., 2017*). More recently, the affective dimension of pain has received more attention, resulting in a multivariate pattern predictive of negative affect ratings to thermal pain, referred to here as the thermal pain aversive signature (TPAS; *Čeko et al., 2022*). Finally, a signature was elaborated to characterise the neuronal representations associated with the valuation of pain (PVP) in the context of a decision task involving a cost-benefit analysis of future pain against a monetary reward (*Coll et al., 2022*). Taken together, those signatures have contributed to improve our understanding of the neurobiological mechanisms of pain, as reflected in self-report or explicit decision-making.

Facial expression has been used as a reliable behavioural measure of pain across different mammal species (*Dalla Costa et al., 2014*; *Craig, 1992*; *Evangelista et al., 2019*; *Langford et al., 2010*; *Sotocinal et al., 2011*), but few studies have investigated the brain mechanisms associated with the spontaneous non-verbal communication of pain in humans (*Kunz et al., 2011*; *Kunz et al., 2020*). As an automatic behavioral manifestation, pain facial expression is considered to be an indicator of activity in nociceptive systems, and to reflect perceptual and affective-evaluative processes. Here, we assessed the association between pain facial expression and the available pain-relevant brain signatures and we applied multivariate analysis with machine learning techniques to develop a predictive brain activation model of the facial responses to pain.

## Results and discussion

The facial action coding system (FACS; *Ekman and Friesen, 1978*) was used to quantify the facial expression of pain in healthy participants while brain responses evoked by brief moderately painful heat stimulation were recorded using fMRI. For each trial, the intensity and the frequency of pain-related action units were scored and combined into a FACS composite score (Materials and methods). The association with the NPS, the SIIPS-1, the PVP, and the TPAS was assessed across the whole brain using the correlation between the FACS scores and the dot product computed between each signature and the activation maps for each individual trial.

Pain facial expression was not significantly associated with NPS expression (*pearson-r*=.06; p=0.20; 95% CI = [–0.03, 0.14]), TPAS expression (*pearson-r*=0.05; p=0.26; 95% CI = [–0.04, 0.14]), PVP expression (*pearson-r*=0.02; p=0.67) and SIIPS-1 expression (*pearson-r*=0.07; p=0.10; 95% CI = [–0.01, 0.16]). These low values indicate that the available pain-relevant brain signatures show poor sensitivity to the facial expression of pain. This motivated the development of a new multivariate brain pattern to predict pain expression.

We used a multivariate approach at the voxel level across the whole brain to develop the FEPS (see Materials and methods). A LASSO principal component regression was applied to predict the FACS composite scores from the trial-by-trial fMRI activation maps. The FEPS was able to predict the FACS composite scores with a performance significantly above chance level (averaged cross-validated prediction across 10 folds: *pearson-r*=0.54 ± 0.10 (95% CI = [0.39; 0.64]); $R^2$=0.22 ± 0.08 (95% CI = [–0.09; 0.33]); RMSE = 0.99 ± 0.08 (95% CI = [0.88; 1.10]); p<0.001 compared to a permuted distribution; *Figure 1-AB*). These results indicate that we were able to develop a multivariate brain pattern accounting for some variance in the facial responses related to pain.

The distributed pattern of activity predicting pain expression was projected back on the brain to examine the spatial distribution of higher weights contributing to the prediction (*Figure 1C*). Positive weight clusters were found in the primary motor cortex (M1; bilateral), the frontal pole, the right posterior parietal cortex, and the dorsal part of the parietal operculum, adjacent to the secondary somatosensory cortex (S2) (*Supplementary file 1A*). These regions are, respectively, associated with motor control, reward and affective value, attentional processes, and nociceptive pain processing (*Price, 2000*; *Rushworth et al., 2011*; *Shackman et al., 2011*). Regions showing negative weights included the dorsolateral PFC (dlPFC), the ventrolateral PFC (vlPFC), the mid-cingulate cortex (MCC), the subgenual ACC, the ventral part of the parietal operculum, the precuneus, and the vmPFC (*Supplementary file 1B*). Negative weights imply that increased activity in those regions is associated with decreased facial response, consistent with a role in the inhibition of pain expression (*Kunz et al., 2011*). The contribution of the dlPFC and the vlPFC to the model's prediction aligns with the role of these regions in inhibitory control and cognitive regulation, respectively (*Goldin et al., 2008*).

A supplementary analysis was conducted to evaluate whether the activity pattern in the primary motor cortex (M1) alone could be sufficient for predicting facial expressions. The choice of this particular region was informed by prior research indicating that M1 presented the strongest correlation with the facial expression of pain (*Kunz et al., 2011*). If the facial expression of pain primarily reflected a motor component without providing substantial insights into the pain experience, then the activity of the motor cortex alone should have been equally effective as the activity of the whole brain in predicting the FACS scores. This M1 model did lead to a significant prediction of pain facial expression (see *Figure 1—figure supplement 2*), but the whole brain model was significantly better (t(532) = 2.73, p=0.003).

These results are consistent with the distributed nature of brain activity associated with the production and regulation of pain facial expression reflecting in part the ascending nociceptive response and the ensuing affective processes, as well as top-down socio-affective regulation underlying the implementation of learned display rules (*Karmann et al., 2016*).

This study was not designed to assess the specificity of the FEPS against other aversive states but the warm stimulation condition allowed us to compare the FEPS response to a painful vs a non-painful thermal stimulation. The FEPS expression score was computed on the activation maps of the warm trials and compared to the pain trials, with and without pain facial expression (*Figure 1—figure supplement 3B*). The results indicated higher expression scores in the painful condition than in the warm condition (Pain - Warm contrast: EMM = 0.24 ± 0.02, t(1034)=10.33, p<0.0001, 95% CI = [0.20, 0.29], Cohen's *d*=0.63). Given the intra-individual and inter-individual variability in facial responses to pain, we repeated this comparison, stratifying the trials within the pain condition based on FACS scores: null FACS scores (FACS = 0) and non-null FACS scores (FACS >0). The FEPS scores were larger in the pain condition where facial responses were displayed, compared to both the pain condition without facial expression and the warm condition (Pain$_{FACS>0}$ - Pain$_{FACS=0}$ contrast: EMM = 0.58 ± 0.03, t(1065)=17.19, p<0.0001, 95% CI = [0.50, 0.66], Cohen's *d*=1.69; Pain$_{FACS=0}$ - Warm contrast: EMM = –0.06 ± 0.03, t(1057)=–2.24, p=0.07, 95% CI = [–0.13, 0.003], Cohen's *d*=–0.18; Pain$_{FACS>0}$ - Warm contrast: EMM = 0.52 ± 0.03, t(1055)=19.64, p<0.0001, 95% CI = [0.46, 0.58], Cohen's *d*=1.52). Note that the comparison of Pain$_{FACS>0}$ vs Pain$_{FACS=0}$ is redundant with the regression approach used as the

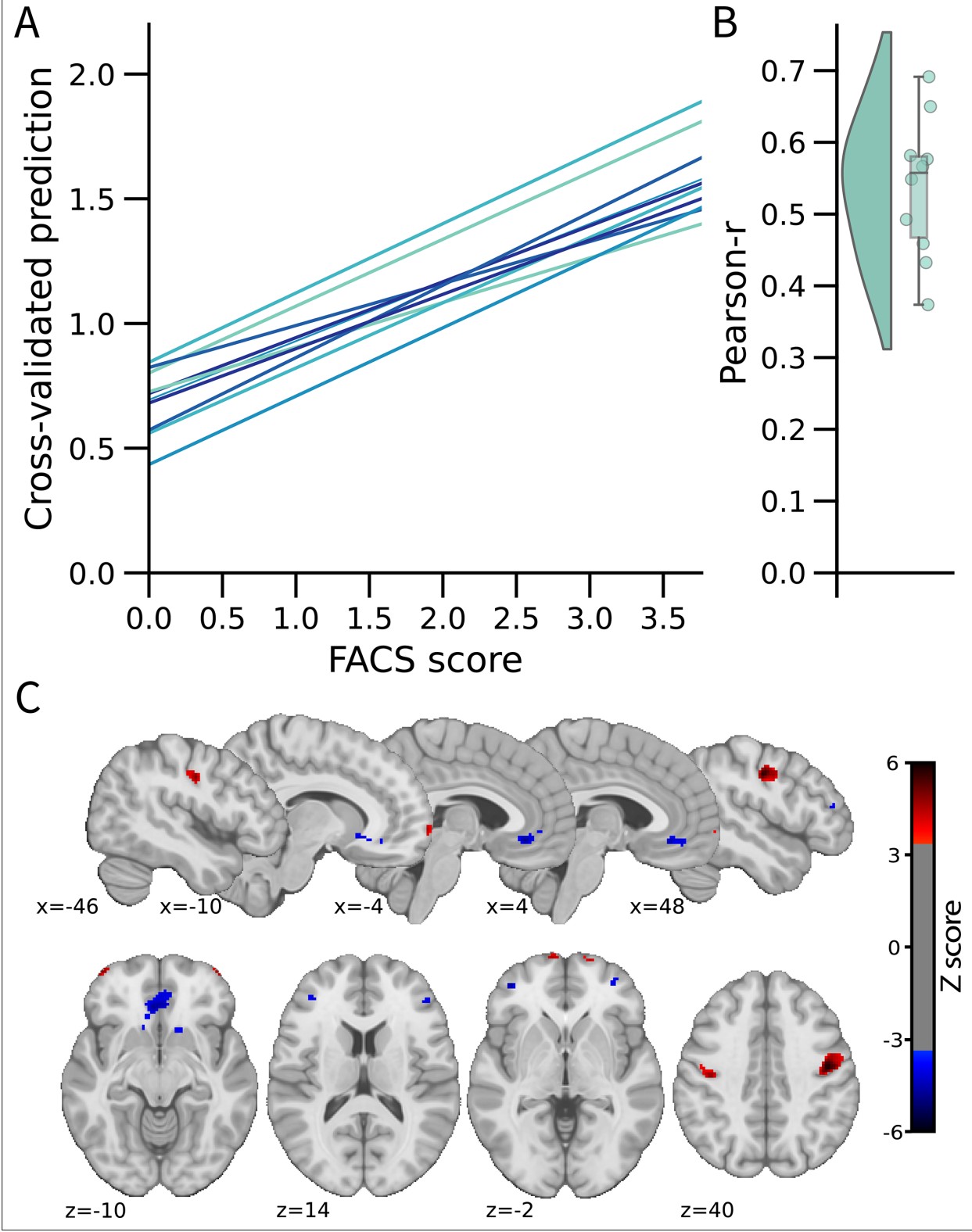

**Figure 1.** Facial expression of pain signature (FEPS): a brain signature of the facial expression of pain. (**A**) Relationship between the actual and the predicted facial action coding system (FACS) composite scores for each cross-validation fold. (**B**) Distribution of the Pearson's r scores across the cross-validation folds. (**C**) Predictive weight map of pain expression thresholded at FDR *corrected* $q<0.05$ using bootstrap tests performed with 5000 samples (with replacement). The thresholded map is shown for visualization and interpretation purposes only, although the prediction was made using voxel weights across the whole brain. MNI coordinates of the clusters with the related z-score can be found in ***Supplementary file 1A and B***. The colour bar

*Figure 1 continued on next page*

*Figure 1 continued*

represents the z-scored regression weights reflecting the positive and negative association with the magnitude of the FACS composite score of pain expression.

The online version of this article includes the following figure supplement(s) for figure 1:

**Figure supplement 1.** Behavioral scores across trials.

**Figure supplement 2.** Predictive performance of the M1-based model.

**Figure supplement 3.** Facial expression of pain signature (FEPS) pattern expression and pain facial expression.

primary analysis model (*Figure 1*) and should not be considered as additional evidence. The observation that the Pain$_{FACS=0}$ trials did not differ significantly from the Warm trials and that both conditions showed a mean score close to 0 (*Figure 1—figure supplement 3B*) indicate that the FEPS does not respond to innocuous thermal stimuli and only responds to noxious heat when a facial expression is produced.

Several regions identified in the FEPS have also been reported in other pain-related brain signatures. Regions predictive of pain facial expression and pain intensity (NPS and SIIPS-1) include S2, the vmPFC, and the precuneus. The vlPFC is a region that does not receive direct spino-thalamo-cortical nociceptive afferents, and was reported both in the FEPS and in the SIIPS-1. Overlap between the FEPS and the PVP (pain value pattern) includes regions associated with reward and affect (i.e. OFC). Finally, the primary motor cortex, and S2 were also reported as contributing regions in the TPAS. The spatial comparison showing some common regions across these pain-relevant signatures suggests possible shared features with the FEPS.

We computed the cosine similarity between the FEPS and other pain-related brain signatures to further examine the shared and specific representations between those predictive patterns (see Materials and methods). Cosine similarity ranging from 0.00 to 0.10 was found between the FEPS and the other pain-related brain signatures reflecting the overall low similarity between the signatures at the whole-brain level (*Figure 2A*). The highest similarity value with the FEPS was found for the SIIPS-1, consistent with the notion that the facial expression of pain may reflect, at least partly, changes in brain responses associated with spontaneous fluctuations in pain experience captured by pain ratings. Similarity with the FEPS was further assessed across different cortical networks (*Figure 2B*). The significant positive similarity with the SIIPS-1 at the frontoparietal level and in the default-mode network may suggest common mechanisms in self-representation, prediction, and emotional regulation of the pain experience that would be reflected in both facial expression and subjective reports (*Pan et al., 2018*). Recruitment of the frontoparietal network may also be involved in the conscious representation of the pain context, making nociceptive information available for integration into decision-making processes (*Coll et al., 2022*; *Bastuji et al., 2016*; *Del Cul et al., 2007*; *Zheng et al., 2020*). The similarity between the FEPS and the SIIPS-1 in the somatomotor network indicates potential overlaps between the sensory aspect of the pain experience captured by the facial expression and the pain intensity ratings. This is consistent with our previous report showing that changes in pain facial expression by the cognitive modulation of perceived pain intensity are correlated to changes in the nociceptive response of the somatosensory cortex (*Kunz et al., 2020*). Finally, the convergent similarities in the limbic network with the PVP is consistent with a key role of affective pain processing influencing facial expression, and perceived pain value (*Garcia-Larrea and Peyron, 2013*; *Roy et al., 2009*).

In research and clinical practice, verbal reports of perceived pain intensity are considered to be the gold standard for measuring pain. Other measures that are often weakly correlated with those subjective reports, like facial expressions of pain, are often considered a less valid metric of the experience of pain even though they provide important complementary information on pain-related processes (*Hadjistavropoulos et al., 2011*). The FEPS was able to predict the magnitude of the facial expression of pain above the chance level. Regions that contribute to the prediction include motor and pain-related areas associated with both sensory and affective processing of pain. Although it shares, to some extent, similar representations with other pain-relevant signatures within various cerebral networks, the FEPS is distinctive from these other signatures. Results of this study provide unique evidence of the complementary information provided by facial expression on pain-related brain processes. Future studies must provide a more comprehensive account of diverse pain manifestations and their related function to better capture the pain phenomenon in its entirety.

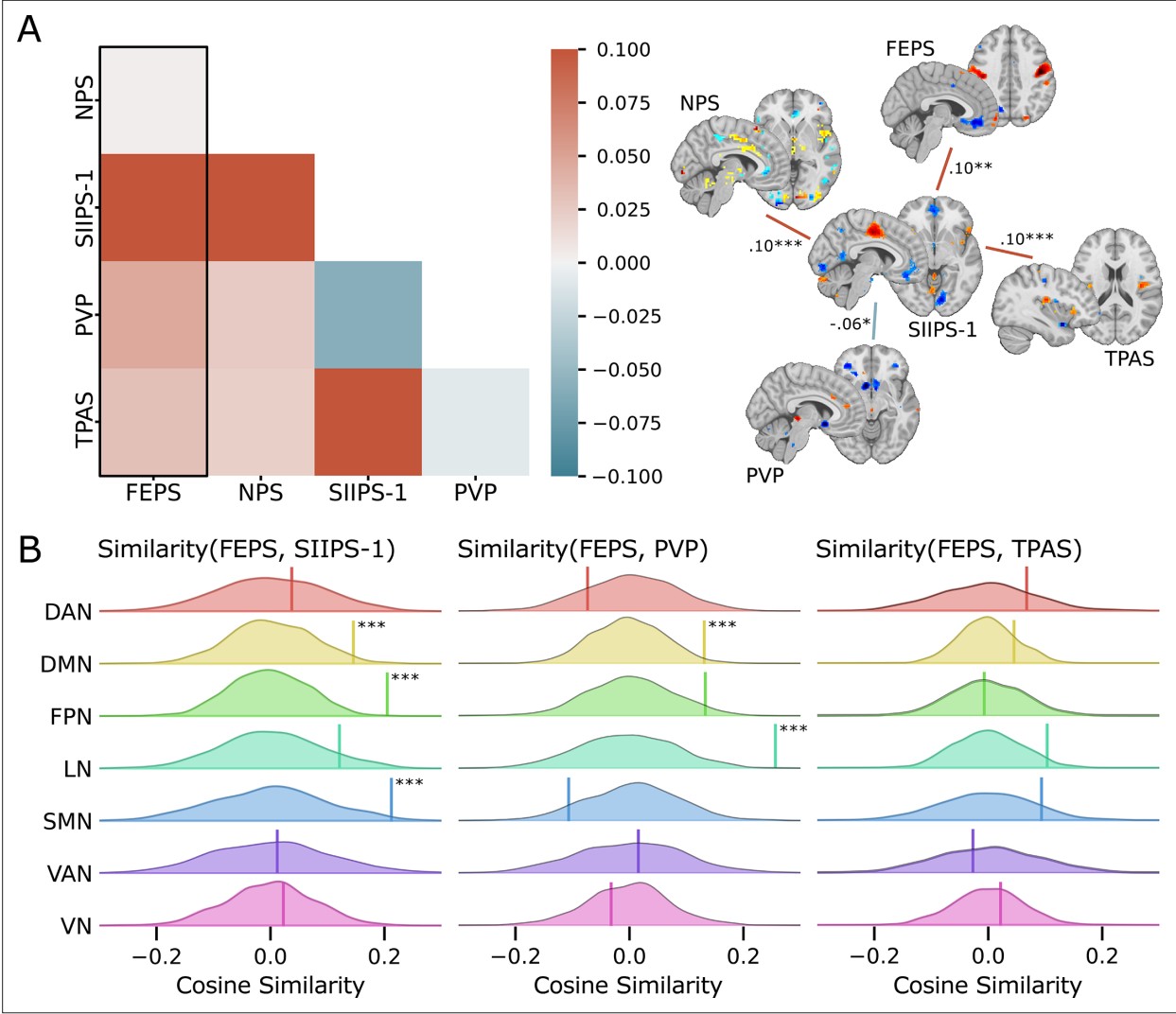

**Figure 2.** Spatial similarity between the facial expression of pain signature (FEPS) and other pain-related signatures. (**A**) Pattern similarity between the FEPS and other pain-related brain signatures using the weights of the full brain patterns. Pattern similarities were computed at the voxel level using the cosine similarity; a value of 1 reflects proportional patterns; a value of 0 reflects orthogonal patterns; a value of –1 reflects patterns of opposite directions. The left panel shows the similarity matrix, and the right panel shows only the significant similarities between the pain-related signatures (*p<0.05; **p<0.01; ***p<0.001). (**B**) Deconstructing the pattern similarity with regards to seven cortical networks as defined in the Yeo atlas[24]: Visual Network (VN); Somatomotor Network (SMN); Dorsal Attention Network (DAN); Ventral Attention Network (VAN); Limbic Network (LN); Frontoparietal Network (FPN); Default Mode Network (DMN). Null distributions computed using permutation tests are shown, and the actual similarity values are represented by the vertical bar. Significant similarity values were found in the FPN (similarity = 0.20; p=0.002), the SMN (similarity = 0.21; p=0.02), and the DMN (similarity = 0.15; p=0.04) for the SIIPS-1, in the LN (similarity = 0.26; p=0.001), and DMN (similarity = 0.13; p=0.03) for the pain value pattern (PVP).

## Limitations

We recognize that our study has several limitations. First, given our limited sample size, further research will be necessary to verify the generalizability of the FEPS across other samples, but also across diverse experimental conditions (e.g. electrical, mechanical, and chemical pain) and populations (e.g. young vs old, chronic pain). Conducting future generalizability studies is crucial to ensure that the FEPS is a valid signature and is not only a result of model overfitting.

Even though the model developed from the entire brain activity could predict pain facial expression scores (FACS scores) beyond chance levels, it is important to highlight its inability to accurately predict the higher facial expression scores. This observation may be explained by the positive asymmetry in the distribution of facial expression scores, despite the log transformation applied to

mitigate the observed skewness in the behavioral data. It is possible that the brain signature of pain facial expression might not adequately capture the heterogeneity of facial expressiveness in the population, especially for highly expressive individuals (*Kunz et al., 2008*). To address this limitation in the generalizability of the model, it could be re-trained with more observations associated with high FACS scores, thereby improving the extraction of predictive features associated with a greater level of facial expressivity. It is also possible that non-linear models might provide a better prediction and/or that the higher pain expressiveness might engage additional brain mechanisms not captured here.

The whole-brain model also overestimated the lowest facial responses with the intercept of the regression lines being systematically greater than 0 (see *Figure 1A*). This means that even when no facial expression was detected, there was still some brain activation matching the FEPS. Again, this may reflect a limitation of the linear model applied here. It is also possible that the FACS method might miss subtle movements of the face and/or that the FEPS captures meaningful variability in the pain-related brain activity below the threshold of facial expression.

It is also essential to assess the specificity of the FEPS in future studies. This involves examining whether the FEPS responds specifically to facial expressions of pain rather than broadly reflecting any facial movements including simulated facial expressions of pain and facial expression of emotions. Despite the similarities between facial expressions of pain and those associated with negative emotions, it is possible to behaviorally distinguish them. This suggests the potential to identify distinct brain patterns predictive of different facial responses (*Simon et al., 2008*). However, to our knowledge, there are no available brain imaging datasets to assess the specificity of the FEPS in that context.

Due to our limited sample size, we were not able to analyze each pain-related facial action unit separately or to explore different combinations. This could be valuable, especially considering the reported differences in the association between diverse pain-related action units and the sensory and affective components of pain (*Kunz et al., 2011*; *Kunz et al., 2020*).

Despite the limitations of this study, the evidence provided by our findings highlights the importance of facial expression as a complementary source of information to assess central nociceptive processes and acute pain. These results should be regarded as a benchmark for future research on non-verbal pain-related manifestations and may provide a foundation for the assessment of brain mechanisms underlying non-verbal communication across domains.

## Materials and methods

### Participants

Secondary analyses of brain imaging data acquired in 34 healthy participants were performed in this study (18 women, 16 men; age mean ± SD = 23.4±2.5 years)(*Kunz et al., 2011*). No participants reported having a history of neurological or psychiatric disease, nor chronic pain. Participants reported not using alcohol nor analgesics for at least 24 hr before the experimental session. All participants provided written informed consent and received monetary compensation for their participation. The information about the video recording of the face was indicated in the consent form but this was not emphasized and it was not mentioned at the time of data acquisition. All procedures were approved by the ethics committee of the Centre de recherche de l'institut universitaire de gériatrie de Montréal.

### Study design

#### Pre-experimental session

Participants were submitted to a pre-experimental session to assess the range of thermal pain sensitivity using a magnitude estimation procedure. All participants included in this study had normal thermal pain sensitivity (see *Kunz et al., 2008* for more details regarding the procedure; *Rainville et al., 1992*). The degree of facial expressiveness was also evaluated (low/nonexpressive (n=13): facial responses in 20% or less of painful trials; expressive (n=21): facial responses in more than 20% of painful trials; *Kunz et al., 2011*). Our sample was representative of the interindividual variability in facial expressivity of pain as reported in the literature (*Kunz et al., 2006*; *Kunz et al., 2008*) and data from all participants were used in the present study.

## Thermal stimuli

Thermal stimulations were induced using a Medoc TSA-2001 thermosensory stimulator with an MRI-compatible 3×3 cm² contact probe (Medoc), positioned at the level of the lower left leg. Thermal stimuli lasted 9 s (2 s ramp up, 5 s plateau at targeted temperature, 2 s *ramp down*) and were followed by an interstimulus interval of 18–25 s. The experiment was programmed using E-prime software (Psychology Software Tools, Pittsburgh, Pennsylvania, United States). A baseline temperature of 38 °C was applied between stimuli for all participants. The target temperatures were determined individually before the MRI scans to induce a warm non-painful sensation in the control condition, and a moderate to strong self-reported pain intensity (targeting 75-80/100 on the pain scale, where 50/100 corresponds to the pain threshold; temperature (mean ± SD)=47.8 ± 0.90 °C). Participants were not aware that the warm and painful temperatures remained constant across trials. The order of the control condition and experimental condition was pseudorandomized. There were eight trials for each experimental condition per run, for a total of 16 trials per condition and a total of 544 trials per condition across all participants (34 participants × 16 pain trials). After each stimulus, participants rated the warm or pain sensation by moving a cursor with an MRI-compatible response key on a computerized visual analog scale.

## Facial expression

The facial expression of the participants was recorded using an MRI-compatible camera (MRC Systems) mounted onto the head coil of the MRI scanner. To be able to quantify facial expressions that occurred during the stimulation, a signal was sent from the stimulator to the sound card to automatically mark the onset of each stimulus on the video recording. Two certified FACS coders evaluated the video recordings to rate the frequency, and intensity (on a 5-point scale) of pain-related action units (AUs; AU4, AU6-7, AU9-10, and AU43) for each trial (see *Kunz et al., 2011* for details about the AUs selection). In the frequency count, an AU was considered as occurring either for the first time or if it was already present, an intensity increase of 2 points was recorded as a new occurrence. Additionally, if there was a distinct interruption followed by a reappearance of the AU for a given time window, it was also added to the frequency count. From the frequency and intensity scores, a composite score (FACS composite score) was computed by taking the product between the mean AU frequency and mean AU intensity values, reflecting pain expression for each trial. A logarithmic transformation was applied in order to normalize the FACS composite scores ($Transformed scores = log (FACS scores + 1)$; skewness = 0.75, kurtosis = –0.84). The transformed FACS composite scores during the painful trials were used as the predictive variable. All results were reported based on the log-transformed scores. To examine whether the facial expression could be confounded with the pain reports, we predicted the trial-by-trial FACS composite scores from the pain ratings using a mixed effect model with the participants as a random effect, and allowing random slopes. Pain ratings were not associated to facial responses ($R^2_{GLMM(m)}$=0.008, $R^2_{GLMM(c)}$=0.74, $\beta$=0.10 ± 0.07, 95% CI = [–0.01; 0.21], $t$(32.57)=1.82, p=0.07, *Supplementary file 1C*). Even if the likelihood and the strength of facial expression of pain generally increase with pain ratings in response to the increase of stimulus intensity, this result is not surprising in the present context where the stimulus intensity is held constant, and spontaneous fluctuations in both facial expression and subjective ratings are observed (see *Kunz et al., 2011* and *Kunz et al., 2018* for a discussion on those results).

To test for a possible habituation or sensitization effect on the facial expressivity throughout the experiment, a mixed effect model was conducted with the trials, the runs, and the interaction between the trials and the runs considered as fixed effects, and the participants as random effects. No evidence of such habituation or sensitization on the log-transformed FACS scores were found in the results ($R^2_{GLMM(m)}$=0.00; $R^2_{GLMM(c)}$=0.71; trial: $\beta$=–0.002 ± 0.02, 95% CI = [–0.03; 0.03], $t$(496.01)=–0.12, p=0.90; run: $\beta$=0.02 ± 0.22, 95% CI = [–0.22; 0.25], $t$(496.16)=0.15, p=0.88; trial × run: $\beta$=0.001 ± 0.02, 95% CI = [–0.05; 0.05], $t$(496.05)=0.05, p=0.96; *Figure 1—figure supplement 1A* and *Supplementary file 1D*). The same analysis conducted on the pain intensity ratings led to the same conclusions ($R^2_{GLMM(m)}$=0.01; $R^2_{GLMM(c)}$=0.49; Trials: $\beta$=0.01 ± 0.21, 95% CI = [–0.40; 0.42], $t$(496.13)=0.04, P=0.97; Runs: $\beta$=1.98 ± 1.47, 95% CI = [–0.91; 4.87], $t$(496.47)=1.35, p=0.18; Trials × Runs: $\beta$=–0.03 ± 0.29, 95% CI = [–0.61; 0.54], $t$(496.21)=–0.12, p=0.91; *Figure 1—figure supplement 1C* and *Supplementary file 1E*).

## Anatomical and functional acquisition

MRI images were acquired using a 12-channel head coil 3T Siemens TRIO scanner. T1-weighted structural data were collected using a MP-RAGE sequence (TR = 2300 ms, TE = 2.91 ms, flip angle = 9°, FOV = 256 mm, matrix size = 240 × 256, voxel size = 1 × 1 × 1.2 mm, 170 whole-brain volumes). Functional data were acquired using an EPI T2*-weighted sequence (TR = 3000 ms, TE = 30 ms, flip angle = 90°, FOV = 220 × 220 mm$^2$, voxel size = 3.44 × 3.44 × 3.40 mm, 40 interleaved axial slices).

## Preprocessing of fMRI data

The fMRI data were preprocessed using SPM8 (Statistical Parametric Mapping, Version 8; Wellcome Department of Imaging Neuroscience, London, United Kingdom) on MATLAB 7.4 (The MathWorks Inc, Natick, Massachusetts, United States). Preprocessing steps included a slice-timing correction, a correction for head movements, and co-registration between functional and anatomical images for each participant. Functional images were normalized into the MNI space. A spatial smoothing procedure (6 mm FWHM Gaussian kernel) and a high pass filter (128 s) were also applied.

BOLD signal was modeled using a canonical hemodynamic response function. First-level analyses were computed using the GLM to obtain a pain activation map for each trial using SPM8. Additionally, the six movement parameters and averaged signals from the white matter and the cerebrospinal fluid were included as nuisance regressors. Eleven trials were discarded due to excessive movements in the painful condition, and eight trials were excluded in the warm condition, resulting in a total of 533 and 536 activation maps for the pai and the warm conditions, respectively. The activation maps of individual painful trials were used to develop theFEPS.

## Analyses

### Association between the facial expression of pain and pain-related brain signatures

The dot product between the neurologic pain signature (NPS; *Wager et al., 2013*), the stimulus intensity independent pain signature-1 (SIIPS-1; *Woo et al., 2017*), the predictive value of pain (PVP; *Coll et al., 2022*), the Thermal Pain Aversive Signature (TPAS; *Čeko et al., 2022*), and the trial-by-trial activation maps was computed to derive a measure of similarity (pattern expression) between the unthresholded signatures and the activation maps. These scalar values were then correlated with the FACS composite scores to assess the association between the facial expression of pain and the NPS, the SIIPS-1, the PVP, and the TPAS, separately. Pearson-r correlation coefficients and p-values are reported.

### Multivariate pattern analysis

We applied a least absolute shrinkage and selection operator principal component regression (LASSO-PCR) with a 10-fold cross-validation procedure for multivariate pattern analysis (*Wager et al., 2011*) using scikit-learn implementation (*Pedregosa et al., 2011*). The algorithm was trained on ~70% of the data and tested on the remaining ~30%, and the LASSO alpha hyperparameter was set at 1.0. The analyses were performed using the trial-by-trial activation maps as input to the model and the participants as a grouping factor (i.e. data from a given participant could only be either in the training set or the testing set for a given cross-validation fold). This procedure was used to predict FACS composite scores from activation maps. The performance of each regression model was evaluated using Pearson's correlation coefficient (*pearson-r*), coefficient of determination (R$^2$; computed using scikit-learn), and root mean square error (RMSE). The scikit-learn's implementation of the coefficient of determination was used, allowing negative values if the model showed worse performance compared to an unfitted model (i.e. a horizontal line). The averaged performance metrics across folds are reported for each analysis. To test if the models performed significantly above chance, permutation tests were computed using 5000 iterations, leading to a p-value corresponding to the probability that the R$^2$ between the observed and predicted FACS scores would be obtained by chance. A bootstrap resampling procedure was also performed to evaluate the stability of the voxel contribution to the model performance, and to derive confidence intervals for the performance metrics. This procedure consists of randomly sampling 5000 times the dataset with replacement. The resulting samples contain the same number of observations as the dataset. The LASSO-PCR procedure as described

above is then applied on each sample. Z-scores and p-values are calculated on the overall regression coefficients.

This analysis procedure was first applied at the whole brain level. It was repeated using a spatial mask including only the precentral region bilaterally. This mask was derived from the Oxford-Harvard Cortical Atlas (*Caviness et al., 1996*). This secondary analysis was conducted to verify if the pattern of activity within the primary motor cortex (M1) might be sufficient to predict facial expression. The performance between the model based on the primary motor cortex activity and the whole-brain model was compared using a corrected resampled t-test (*Nadeau and Bengio, 1999*). To ensure that the whole brain model prediction of the facial responses was not confounded with the pain ratings, we predicted the facial composite scores from the FEPS pattern expression scores (i.e. the dot product between the trial-by-trial activation maps and the unthresholded FEPS signature), and included the trial-by-trial pain ratings using a mixed effect model including the participants as a random effect, and allowing the slopes to vary. Variance in the facial composite scores was significantly explained by the FEPS pattern expression scores ($R^2_{GLMM(m)}$=0.40; $R^2_{GLMM(c)}$=0.69; FEPS scores: $\beta$=0.62 ± 0.04, 95% CI = [0.54; 0.70], $t$(431.79)=14.8, p<0.001; pain ratings: $\beta$=0.02 ± 0.05, 95% CI = [−0.07; 0.11], $t$(30.34)=0.48, p=0.68; FEPS scores × pain ratings: $\beta$=−0.09 ± 0.04, 95% CI = [−0.17; −0.01], $t$(194.89)=−2.25, p=0.03; see *Supplementary file 1F*; also see *Figure 1—figure supplement 3A* for the scatterplot and the regression line between the log-transformed scores and the FEPS pattern expression scores across all data points). These results confirm the prediction of the facial response by the FEPS scores even when pain ratings are included as a predictors in the model. However, a small but significant negative interaction between the FEPS scores and the pain ratings was found. This possible moderator effect indicates that, for a constant stimulus, the positive slope between the FEPS scores and the facial responses is slightly reduced when pain ratings are higher. This may reflect a saturation effect across the two output channels (i.e. verbal reports and facial expressions).

## Response of the FEPS to pain and warm

To test if the FEPS was more activated during the painful condition compared to the warm conditions, the FEPS expression scores were further computed on the warm trials. A linear mixed model was performed to examine the relation between the FEPS scores and the experimental conditions (Warm and Pain), considering those conditions as a fixed categorical effect and the participants as a random effect. Contrasts on the estimated marginal means (least-squares means) were conducted to assess the statistical significance of the difference between the FEPS expression scores on the warm trials and the painful trials. Given that there are also trials where the FACS scores were equal to zero in the painful condition, the same analyses were repeated, this time separating the FEPS expression scores based on whether FACS scores were greater or equal to 0 in the painful condition. The estimated marginal means (EMM) contrast between experimental conditions were reported with the associated *t*-value and *p*-value (corrected for multiple comparisons using the Tukey method).

## Spatial similarity across the FEPS and pain-related brain signatures

Similarity between the FEPS and other pain-related brain signatures was assessed using the cosine similarity computed across all voxels ($Similarity\,(X, Y) = \frac{X \cdot Y}{|X||Y|}$). This metric was computed on the unthresholded NPS, SIIPS-1, PVP, and TPAS maps. To further explore the cortical similarities between the FEPS and other pain-related brain signatures, we also computed the cosine similarity across different cortical networks (i.e. visual, somatomotor, dorsal attention, ventral attention, limbic, frontoparietal, and default mode) (*Yeo et al., 2011*). Permutation tests (n=1000) using generative null models preserving the spatial autocorrelation of a target brain map were used to assess the significance of the similarity between the brain signatures (*Burt et al., 2020*). p-values were calculated from the generated null distributions as the fraction of permuted samples where the absolute cosine similarity was equal, or larger than the cosine similarity obtained on the original signatures.

## Acknowledgements

We wish to thank André Cyr and Carollyn Hurst from the Unité de Neuroimagerie Fonctionnelle for their help with the data acquisition, and Andréanne Proulx, François Paugman, and Pierre Bellec for their insights on the analyses as part of BrainHack School. We also want to acknowledge Compute Canada for the computational resources allocated to conduct the analyses.

## Additional information

### Funding

| Funder | Grant reference number | Author |
|---|---|---|
| Fonds de recherche du Québec – Nature et technologies | | Marie-Eve Picard |
| Institut de Valorisation des Données | | Marie-Eve Picard |
| Natural Sciences and Engineering Research Council of Canada | RGPIN-2018-06799 | Pierre Rainville |
| National Institutes of Health | R01 DA035484 | Tor D Wager |

The funders had no role in study design, data collection and interpretation, or the decision to submit the work for publication.

### Author contributions

Marie-Eve Picard, Conceptualization, Software, Formal analysis, Visualization, Writing – original draft, Writing – review and editing; Miriam Kunz, Conceptualization, Data curation, Methodology; Jen-I Chen, Data curation, Formal analysis, Methodology; Michel-Pierre Coll, Software, Validation, Visualization; Etienne Vachon-Presseau, Conceptualization, Validation, Methodology; Tor D Wager, Conceptualization, Funding acquisition, Validation; Pierre Rainville, Conceptualization, Supervision, Funding acquisition, Methodology, Writing – original draft, Writing – review and editing

### Author ORCIDs

Marie-Eve Picard ● https://orcid.org/0009-0001-2412-7829
Michel-Pierre Coll ● https://orcid.org/0000-0002-1475-5522
Etienne Vachon-Presseau ● https://orcid.org/0000-0002-8681-3154
Pierre Rainville ● https://orcid.org/0000-0001-9801-757X

### Ethics

All participants provided written informed consent and received monetary compensation for their participation. The information about the video recording of the face was indicated in the consent form but this was not emphasized and it was not mentioned at the time of data acquisition. All procedures were approved by the ethics committee of the Centre de recherche de l'institut universitaire de gériatrie de Montréal (Comité d'éthique de la recherche vieillissement-neuroimagerie - R9; reference number CMER-RNQ 14-15-003).

Reviewer #2 (Public review): https://doi.org/10.7554/eLife.87962.3.sa1
Reviewer #3 (Public review): https://doi.org/10.7554/eLife.87962.3.sa2
Author response https://doi.org/10.7554/eLife.87962.3.sa3

## Additional files

### Supplementary files

• Supplementary file 1. Supplementary tables. (**A**) Peak regions with positive weights contributing to the prediction of the facial expression scores. (**B**) Peak regions with negative weights contributing to the prediction of the facial expression scores. (**C**) Mixed-effect model

results for the effect of pain ratings on the logarithmic transformed facial action coding system (FACS) scores. (**D**) Mixed-effect model results for the effect of runs and trials on the logarithmic transformed FACS scores. (**E**) Mixed-effect model results for the effect of runs and trials on the pain ratings. (**F**) Mixed-effect model results for the effect of the facial expression of pain signature (FEPS) expression scores and the pain ratings on the logarithmic transformed FACS scores.

• MDAR checklist

### Data availability

This report is based on the secondary analysis of data previously acquired for another study in our laboratory (see *Kunz et al., 2011*). In that previous study, participants did not provide informed consent for the sharing of individual data. Unthresholded and thresholded FEPS patterns are available on Neurovault (Compact Identifiers: https://identifiers.org/neurovault.collection:13924). The SIIPS-1, the PVP and the TPAS are available on Github (https://github.com/canlab/Neuroimaging_Pattern_Masks, *Petre and Wager, 2024*). We have had access to the NPS through TDW. To have access to it contact Tor.D.Wager@Dartmouth.edu. All the related information can also be found on this website https://sites.google.com/dartmouth.edu/canlab-brainpatterns/multivariate-brain-signatures/2013-nps. Furthermore, all custom Python and R scripts used to produce the analyses are available at https://github.com/me-pic/picard_feps_2023 (copy archived at *Picard, 2024*) under the Apache-2.0 license. The code used for the bootstrap analyses was adapted from https://github.com/mpcoll/coll_painvalue_2021 (*Coll, 2022*).

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
