## [Editor Report · eLife assessment]

Picard et al. propose a Facial Expression Pain Signature (FEPS) derived from functional magnetic resonance imaging (fMRI) data to predict facial expressions associated with painful heat stimulation. This **important** work advances our understanding of the brain mechanisms associated with facial expressions of pain. It provides **solid** evidence that facial expressions of pain contain information that is complementary to other pain-related brain processes. The work will be of broad interest to researchers from varied fields ranging from neurosciences to psychology and affective sciences.

---

## [Referee Report · Reviewer #2 (Public review)]

Summary.

The objective of this study was to further our understanding of the brain mechanisms associated with facial expressions of pain. To achieve this, participants' facial expressions and brain activity were recorded while they received noxious heat stimulation. The authors then used a decoding approach to predict facial expressions from functional magnetic resonance imaging (fMRI) data. They found a distinctive brain signature for pain facial expressions (FEPS). This signature had minimal overlap with brain signatures reflecting other components of pain phenomenology, such as signatures reflecting subjective pain intensity or negative effects.

Strength.

The authors used a rigorous approach involving multivariate brain decoding to predict the occurrence and intensity of pain facial expressions during noxious heat stimulation. The analyses are solid and well-conducted. This is an important study of fundamental and clinical relevance.

Weakness.

Despite those major strengths, the main weakness of the study is that the design and analyses do not allow us to know if the FEPS is really specific to pain expressions. Based on the analysis, it is possible to conclude that this brain signature is present when a participant is in a state of pain and displays a facial expression. However, it is possible that it would also be present when a participant experiences (another) negative state and displays (another) facial expression. It will be important, in future work, to investigate the specificity of this brain signature.

---

## [Referee Report · Reviewer #3 (Public review)]

In this manuscript, Picard et al. propose a Facial Expression Pain Signature (FEPS) as a distinctive marker of pain processing in the brain. Specifically, they attempt to use functional magnetic resonance imaging (fMRI) data to predict facial expressions associated with painful heat stimulation.

The main strengths of the manuscript are that it is built on an extensive foundation of work from the research group, and that experience can be observed in the analysis of fMRI data and the development of the machine learning model. Additionally, it provides a comparative account of the similarities of the FEPS with other proposed pain signatures. The main weaknesses of the manuscript are the absence of a proper control condition to assess the specificity of the facial pain expressions, as well as several limitations in the experimental setup.

I believe that the authors partially succeed in their aims, as described in the introduction, which are to assess the association between pain facial expression and existing pain-relevant brain signatures, and to develop a predictive brain activation model of the facial responses to painful thermal stimulation. However, they list several limitations in the study that should be addressed in future research in order to establish whether FEPS truly conveys distinctive information about the brain response to nociceptive stimuli.

---

## [Author Response]

The following is the authors’ response to the original reviews.

Summary of the changes

Changes in the manuscript were made to clarify some ambiguities raised by the reviewers and to improve the report following their recommendations. A summary of the main changes is listed below:

- The title was changed to better reflect the results of this study - Re-training the model on log transformed FACS scores.

- Testing the specificity of the FEPS to facial expression of pain within this experimental setup by comparing it to the activation maps obtained from the Warm stimulation condition.

- Testing for sensitization/habituation of the behavioral measures (FACS scores and pain ratings).

- Adding a section in the discussion to better address the limitations of this study and provide potential directions for future studies.

Other changes target areas where the original manuscript may have been ambiguous or lacked precision. To address these concerns, additional details have been incorporated, and certain terms have been revised to ensure a more precise and transparent presentation of the information.

**Public Reviews:**

**Reviewer #1 (Public Review):**
Picard et al. report a novel neural signature of facial expressions of pain. In other words, they provide evidence that a specific set of brain activations, as measured by means of functional magnetic resonance imaging (fMRI), can tell us when someone is expressing pain via a concerted activation of distinctive facial muscles. They demonstrate that this signature provides a better characterization of this pain behaviour when compared with other signatures of pain reported by past research. The Facial Expression of Pain Signature (FEPS) thus enriches this collection and, if further validated, may allow scientists to identify the neural structures subserving important non-verbal pain behaviour. I have, however, some reservations about the strength of the evidence, relating to insufficient characterization of the underlying processes involved.

We are thankful for the summary of our work. We are hopeful that the modifications made in the latest version effectively address these concerns. The changes are outlined in the summary above, and detailed in the following point-by-point response.

Strengths:The study relies on a robust machine-learning approach, able to capitalise on the multivariate nature of the fMRI data, an approach pioneered in the field of pain by one of the authors (Dr. Tor Wager). This paper extends Wager's and other colleagues' work attempting to identify specific combinations of brain structures subserving different aspects of the pain experience while examining the extent of similarity/dissimilarity with the other signatures. In doing so, the study provides further methodological insight into fine-grained network characterization that may inspire future work beyond this specific field.

We are thankful for the positive comments.

Weaknesses:The main weakness concerns the lack of a targeted experimental design aimed to dissect the shared variance explained by activations both specific to facial expressions and to pain reports. In particular, I believe that two elements would have significantly increased the robustness of the findings:(1) Control conditions for both the facial expressions and the sensory input. An efficient signature should not be predictive of neutral and emotional facial expressions (e.g., disgust) other than pain expressions, as well as it should not be predictive of sensations originating from innocuous warm stimulation or other unpleasant but non-painful stimulation.

We do recognize the lack of specificity testing for the FEPS, especially towards negative emotional facial expressions. This would be relevant to test given the behavioural overlap between the facial expressions of pain and disgust, fear, anger, and sadness (Kunz
et
al.,
2013; Williams,
2003). The experimental design used in this study did not include other negative states. However, we fully support the necessity of collecting data throughout those conditions, and we believe that the present study highlights the importance of such a demonstration. Future research should involve recording facial expressions while exposing participants to stimuli that elicit a range of negative emotions but, to our knowledge, such combination of fMRI and behavioural data is currently unavailable. As raised by the reviewer, this approach would allow us to assess the specificity of the FEPS to the facial expression evoked by pain compared to different affective states. We would like to emphasise that specificity and generalizability testing is a massive amount of work, requiring multiple studies to address comprehensively. A Limitations paragraph addressing this research direction has been added to the Discussion. A conclusion was added to the abstract as follows: “Future studies should explore other pain-relevant manifestations and assess the specificity of the FEPS against other types of aversive or emotional states.”

(2) Graded intensity of the sensory stimulation: different intensities of the thermal stimulation would have caused a graded facial expression (from neutral to pain) and graded verbal reports (from no pain to strong pain), thus offering a sensitive characterisation of the signal associated with this condition (and the warm control condition).However, these conditions are missing from the current design, and therefore we cannot make a strong conclusion about the generalisability of the signature (regardless of whether it can predict better than other signatures - which may/may not suffer from similar or other methodological issues - another potential interesting scientific question!). The authors seem to work on the assumption that the trials where warm stimulation was delivered are of no use. I beg to disagree. As per my previous comment, warm trials (and associated neutral expressions) could be incorporated into the statistical model to increase the classification sensitivity and precision of the FEPS decoding.

The experience of pain can fluctuate for a fixed intensity or after controlling statistically for the intensity of the stimulation (Woo et al., 2017). Consistent with this, the current study focused on spontaneous facial expression in response to noxious thermal stimuli delivered at a constant intensity that produced moderate to strong pain in every participant. As the reviewer points out, this does not allow us to characterise and compare the stimulus-response function of facial expression and pain ratings. The advantage of the approach adopted is to maximise the number of trials where facial expression is more likely to occur, while ensuring that changes in facial expression and pain ratings are not confounded with changes in stimulus intensity. The manuscript has been revised to clarify that point. However, we do agree that it would be interesting to conduct more studies focusing on facial expression in response to a range of stimulus intensities. This discussion has been added to the Limitations paragraph.

Furthermore, following the reviewer’s suggestion, we performed complementary analyses on the warm trials in the proposed revisions. The dot product (FEPS scores) between the FEPS and the activation maps associated with the warm condition was computed. A linear mixed model was conducted to investigate the association between FEPS scores and the experimental condition (warm vs pain). The trials in the pain condition were divided into two conditions: null FACS scores (painful trials with no facial response; FACS scores = 0) and non-null FACS scores (painful trials with a facial response; FACS > 0). The details of this analysis have been added to the manuscript (see Response of the FEPS to pain and warm section in the Methods; lines 427 to 439) as well as the corresponding results (see Results and Discussion; lines 138 to 158). The FEPS scores were larger in the pain condition where a facial response was expressed, compared to both the pain condition without facial expression and the warm condition. These results confirmed the sensitivity of the FEPS to facial expression of pain.

**Reviewer #2 (Public Review):**
Summary:The objective of this study was to further our understanding of the brain mechanisms associated with facial expressions of pain. To achieve this, participants' facial expressions and brain activity were recorded while they received noxious heat stimulation. The authors then used a decoding approach to predict facial expressions from functional magnetic resonance imaging (fMRI) data. They found a distinctive brain signature for pain facial expressions. This signature had minimal overlap with brain signatures reflecting other components of pain phenomenology, such as signatures reflecting subjective pain intensity or negative effects.

We appreciate this concise and accurate summary of our study.

Strength:The manuscript is clearly written. The authors used a rigorous approach involving multivariate brain decoding to predict the occurrence and intensity of pain facial expressions during noxious heat stimulation. The analyses seem solid and well-conducted. I think that this is an important study of fundamental and clinical relevance.Weaknesses:Despite those major strengths, I felt that the authors did not suffciently explain their own interpretation of the significance of the findings. What does it mean, according to them, that the brain signature associated with facial expressions of pain shows a minimal overlap with other pain-related brain signatures?

We express our sincere gratitude for the valuable insights and constructive comments on the strengths and weaknesses of the current study. We thank reviewer 2 for the encouragement to reinforce our interpretation of the significance of the findings, while acknowledging the limitations raised by the three reviewers.

A few questions also arose during my reading.Question 1: Is the FEPS really specific to pain expressions? Is it possible that the signature includes a facial expression signal that would be shared with facial expressions of other emotions, especially since it involves socio-affective regulation processes? Perhaps this question should be discussed as a limit of the study?

We acknowledge this limitation as outlined in response to Reviewer #1. We have incorporated a Limitations paragraph to provide a more in-depth discussion of this limitation and to explore potential future avenues (lines 225 to 268). Again, please note that the demonstration of specificity is an incremental process that requires a systematic comparison with other conditions where facial expressions are produced without pain. A concluding sentence was added to the abstract to encourage specificity testing in future studies. as indicated above.

Question 2: All AUs are combined together in a composite score for the regression. Given that the authors have other work showing that different AUs may be associated with different components of pain (affective vs. sensory), is it possible that combining all AUs together has decreased the correlation with other pain signatures? Or that the FEPS actually reflects multiple independent signatures?

The question raised is consistent with the work of Kunz, Lautenbacher, LeBlanc and Rainville (2012), and Kunz, Chen and Rainville (2020). In the current study, the pain-relevant action units were combined in order to increase the number of trials where a facial response to pain was expressed, thus enhancing the robustness of our analyses. Given the limited sample size, our current dataset is unfortunately insufficient to perform such analysis as there would not be enough trials to look at the action units separately or in subgroups. While the approach of combining the different AUs has proven to be valid and useful, we recognize the value of investigating potential independent signatures associated with the different AUs within the FEPS, and examining whether those signatures can lead to more similar patterns compared to previously developed pain signatures. This discussion has been included in the Limitations paragraph in the Discussion (lines 225 to 268).

Question 3: Is facial expressivity constant throughout the experiment? Is it possible that the expressivity changes between the beginning and the end of the experiment? For instance, if there is a habituation, or if the participant is less surprised by the pain, or in contrast if they get tired by the end of the experiment and do not inhibit their expression as much as they did at the beginning. If facial expressivity changes, this could perhaps affect the correlation with the pain ratings and/or with the brain signatures; perhaps time (trial number) could be added as one of the variables in the model to address this question.

The concern raised by the reviewer is legitimate. We conducted a mixed-effects model to assess the impact of successive trials and runs on facial expressivity. Results indicate that the FACS scores did not change significantly throughout the experiment, suggesting no notable effect of habituation or sensitization on the facial expressivity in our study. Details about the analysis and the results have been added to the Facial Expression section in the Methods (lines 335 to 346).

**Reviewer #3 (Public Review):**
In this manuscript, Picard et al. propose a Facial Expression Pain Signature (FEPS) as a distinctive marker of pain processing in the brain. Specifically, they attempt to use functional magnetic resonance imaging (fMRI) data to predict facial expressions associated with painful heat stimulation. The main strengths of the manuscript are that it is built on an extensive foundation of work from the research group, and that experience can be observed in the analysis of fMRI data and the development of the machine learning model. Additionally, it provides a comparative account of the similarities of the FEPS with other proposed pain signatures. The main weaknesses of the manuscript are the absence of a proper control condition to assess the specificity of the facial pain expressions, a few relevant omissions in the methodology regarding the original analysis of the data and its purpose, and a biased interpretation of the results.I believe that the authors partially succeed in their aims, as described in the introduction, which are to assess the association between pain facial expression and existing pain-relevant brain signatures, and to develop a predictive brain activation model of the facial responses to painful thermal stimulation. However, I believe that there is a clear difference between those aims and the claim of the title, and that the interpretation of the results needs to be more rigorous.

We wish to express our appreciation for the insightful and constructive critique provided. The limitation pertaining to the absence of specificity testing had been addressed in response to Reviewer #1, and it has been incorporated into the manuscript (lines 251 to 258).

The commentary made by Reviewer #3 has drawn our attention to a critical concern, namely the potential misalignment between the study findings and our original title. Consequently, we have changed the title to “A distributed brain response predicting the facial expression of acute nociceptive pain”. We also revised the interpretation of the results in the discussion section and we have added a section on limitations.

**Recommendations for the Authors:**

**Reviewer #1 (Recommendations For The Authors):**
I hope the following comments will be useful to improve the manuscript.AbstractI felt the abstract could be more clear in terms of experimental or scientific questions, hypotheses/expectations, and findings. I also feel the abstract should briefly support the conclusive claim ("is better than...": how better? Or according to what criterion? This may be more relevant than the final conclusive general sentence that does not specifically address the significance of the findings).

The abstract was revised to reinforce the functional perspective adopted to interpret brain activity produced by noxious stimuli and predicting various pain-relevant manifestations. We also mention explicitly the other pain-relevant signatures against which the FEPS is compared in this report, and we added a concluding sentence highlighting the importance of assessing the specificity of the FEPS in future studies.

Introduction - background and rationaleI would postpone the discussion around pain signature and anticipate the one about the brain mechanisms of facial expressions of pain. This will allow you to reinforce the logical flow of rationale, literature gap/question, why the problem is important, and study aims. Only then go for a review of relevant literature on signatures before providing a more specific final paragraph about the study-specific questions, expectations, and implementation. At the moment this is limited to a single very descriptive short paragraph at the end of the intro.

The introduction was structured to guide the readers through a comprehensive understanding of different pain neurosignatures. The introduction aimed to establish a robust rationale for the subsequent analyses detailed in the results section. Indeed, the presentation of that literature ensured that the discussion around pain signatures is contextualised within a broader continuous framework. We acknowledge the reviewer’s comment on the limited description of the brain mechanisms of facial expression of pain. However, this was addressed in several previous reports of our laboratory (Kunz et al. 2011; Vachon-Presseau et al. 2016; Kunz, Chen, and Rainville 2020). We have added some more details about the brain mechanisms of facial expression, and highlighted those references in the first paragraph of the introduction.

Methods and Results(1) Was there any indication of power based on the previous work or the other signature papers? If yes, how that would inform the present analysis?

The NPS was trained on 20 participants that experienced 12 trials at each of four different intensities. The assessment of the effect sizes was performed on the Neurological Pain Signature in Han
et
al. (2022). That study revealed a moderate effect size for predicting between-subject pain reports, and a large one for predicting within-subject pain reports. We trained our model on 34 participants that underwent 16 trials. We expected our results to show a smaller effect size as the current experimental design only allowed us to examine spontaneous changes in the facial expression, as noted in the comments made by Reviewer #1. However, the best way to calculate the unbiased effect size of the results presented in the current study would be to test the unchanged model on new independent datasets (see Reddan,
Lindquist,
and
Wager,
2017). Unfortunately, such datasets do not currently exist.

(2) I would clarify to the reader what is meant by normal range of thermal pain and why is this relevant. Also, I did not find data about this assessment nor about the assessment of facial expressiveness (or reference to where it can be found).

We changed this formulation to “All participants included in this study had normal thermal pain sensitivity” and we added a few references. By targeting a healthy population with normal thermal pain sensitivity, our study sought to identify a predictive brain pattern related to facial expression evoked by typical responses to pain that could eventually be generalised to other individuals from the same population. Details about the assessment of facial expressiveness have been added in the appropriate section in the Methods.

(3) That pain ratings are only weakly associated with facial responses is, in its own right, an interesting finding, as a naïve reader would expect the two to be highly positively correlated. I'd suggest discussing this aspect (in reference to previous research) as it is interesting on both theoretical and empirical grounds.

The likelihood and the strength of pain facial expression generally increase with pain ratings in response to acute noxious stimuli of increasing physical intensities, thereby leading to a positive association between the two responses that is driven by the stimulus. However, the poor correlation or the dissociation between facial pain expression and pain rating is a very well known phenomenon that can be demonstrated easily using experimental methods where the stimulus intensity is held constant and spontaneous fluctuations are observed in both facial expression and pain ratings. This result was not discussed in the current manuscript as it was already addressed in the work of Kunz et al. (2011) and Kunz, Karos and Vervoot (2018). We added the references to these studies in the revised manuscript (lines 330 to 334).

(4) It may be worth having CIs throughout the whole set of analyses.

Thanks for the suggestions, this was an oversight. The confidence intervals have been added in the manuscript where applicable.

(5) I would clarify if there are two measures of the brain signature: dot-product and activation map. Relatedly, I cannot find where the authors explained what "FEPS pattern expression scores". Can the authors please clarify?

The clarification has been added in the manuscript (lines 413 to 414).

(6) There seems to be the assumption that the relationship between pain-relevant brain signatures and facial expressions of pain would be parametric and linear. However, this might not hold true. Did the authors test these assumptions?

We indeed decided to use a linear regression technique (i.e. LASSO regression) to model the association between the brain activity and the facial expression of pain. The algorithm choice was mainly based on the simplicity and the interpretability of that approach, and our limited number of observations. The choice was also coherent with previous studies in the domain (e.g. Wager et al., 2011; Wager et al., 2013; Krishnan et al. 2016; Woo et al., 2017). Using a linear model, we were able to predict above chance level the facial expression evoked by pain using the fMRI activation. However, it is legitimate to think that more complex non linear models can better capture the brain patterns predictive of that behavioural manifestation of pain.

(7) Did the authors assess whether the FACS were better to be transformed/normalised? More generally, I would report any data assessment/transformation that has not been reported.

Thank you for this highly relevant suggestion. FACS scores were indeed not normally distributed and the analyses were conducted again to predict the log transformed FACS scores. This transformation was effective to normalize the distribution (skewness = 0.75, kurtosis = -0.84). The predictive model was confirmed on transformed data.

(8) Page 12: I am not clear on whether all the signatures are included in the same model (like a multiple regression) or if separate regressions are calculated per signature. The authors seem to imply that several regressions have been computed (possibly one per comparison with each signature?).

The correlation between the FACS scores and the pain-related signatures was computed separately for each signature. This information has been clarified.

(9) MVPA: See my main comment about warm trials and experimental/statistical design. For example, the LASSO regression model for the pain trials could be compared with a model using warm trials besides (or instead of) the unfitted model. Otherwise, add the warm trials as another predictor or within the subject level in a dummy fixed factor comprising pain and warm trials.

The inclusion of warm trials in the model training would be inconsistent with the goal of the main analysis to predict the facial expression of pain when a noxious pain stimulus is presented. Secondary analyses were conducted to compare the response of the FEPS to the warm trials compared to noxious pain trials. The dot product between the FEPS and the activation maps (FEPS scores) associated with the warm condition was computed. A linear mixed model was conducted to investigate the association between FEPS scores and the experimental condition (warm vs pain). Additional contrasts compared the warm trials with the pain trials with and without pain facial expression. The details of this analysis have been added to the manuscript (see Response of the FEPS to pain and warm in the Methods) as well as the corresponding results (see Results and Discussion).

(10) I would clarify for the reader why the separate M1 analysis has been run. Although obvious, I feel the reader would benefit from the specific hypothesis about this control analysis being spelled out together with the other statistical hypotheses within the statistical design in a more streamlined manner.

We extended the discussion on the rationale of that analysis and its interpretation taking into account the most recent results using the log transformed FACS scores (lines 125 to 133).

(11) The mixed model aimed to assess the relationship between pain ratings FEPS scores and facial scores is a crucial finding. I believe it speaks to the importance of a more complete design, which I already highlighted. I have a couple of technical questions: did the authors assess random slopes too? And, what was the strategy used to determine the random effects structure?

The linear mixed model considered the participants as a random effect, with random intercepts, considering the grouping structure in our data (i.e., each participant completed multiple trials). The reported results in the original manuscript were considering fixed slopes. However, following the reviewer’s comment, we re-computed the mixed linear models allowing the slopes to vary according to the intensity ratings. The results were changed in the manuscript to represent the output of those models.

(12) The text from lines 63 to 67 could go in the methods.

We decided to include those lines within the Result and Discussion section to give the reader more specification about the FACS scores, as this term is subsequently referenced in the following part of the Results and Discussion section. We are concerned that putting this information only in the Methods section would disrupt the reading.

**Reviewer #2 (Recommendations For The Authors):**
p. 4-5. When you report the positive weight clusters, you follow up with a sentence specifying which cognitive processes those brain regions are typically associated with. However, when you report the negative weight clusters, you do not specify the cognitive processes typically associated with those brain areas. I think that providing that information would be helpful to the readers.

Thanks for noticing this omission. The information has been added in the most recent version of the manuscript (lines 119 to 121).

p. 9. You specify that the degree of expressiveness of participants was evaluated. How did you evaluate expressiveness? Did you use this variable in your analyses? Were participants excluded based on their degree of expressiveness?

Details about the assessment of facial expressiveness have been added in the appropriate section in the Methods (lines 285 to 289).

p. 10. You explain that two certified FACS-coders evaluated the video recordings to rate the frequency of AUs. Could you please provide more details about the frequency measure? I think that there are different ways in which this could have been done. For instance, were the videos decomposed into frames, and then the frequency measured by summing the number of frames in which the AU occurred? Or was it "expression-based", so one occurrence of an AU (frequency of 1) would correspond to the whole period between its activation onset and offset? Both ways have pros and cons. For example, if the frequency represents the number of frames, then it controls for the total duration of the AU activation within a trial (pro); but if there were multiple activations/deactivations of the AU within one trial, this will not be controlled for (con). And vice-versa with the second way of calculating frequency.

Details about the frequency scores have been added to the manuscript (lines 315 to 319).

p. 11. When you explained how you calculated the association between the facial expression of pain and pain-related brain signatures, I felt that there was some information missing. Did you use the thresholded maps (available in the published articles), or did you somehow have access to the complete, voxel-by-voxel, raw regression coefficient maps?

The unthresholded maps were used. The information has been clarified in the latest version of the manuscript, as well as the details about the availability of the maps (see Data Availability section at the end of the manuscript).

**Reviewer #3 (Recommendations For The Authors):**
FormatThe authors will notice that many observations about the manuscript are related to missing information and a lack of graphical representations. I believe the topic and the content of the manuscript are too complex to condense into a short report.TitleThe claim of the title is simply not substantiated by the content of the manuscript. Demonstrating that the FEPS is a distinctive (i.e., specific) marker of pain processing requires a substantially different experimental design, with more rigorous controls and a broader set of painful stimulations. The manuscript would benefit from a more accurate title.We agree that the title could better align with our findings. We modified the title accordingly : “A distributed brain response predicting the facial expression of acute nociceptive pain”.AbstractI find it puzzling that the authors claim that there is limited knowledge of the neural correlates of facial expression of pain given what they describe in the first paragraph of the introduction. Besides, they propose to reanalyze a dataset that has been extensively described in Kunz et al. (2011), which is unlikely to provide any new significant information.

We respectfully disagree with that comment. We considered that three articles (i.e., Kunz et al., 2011; Vachon-presseau et al., 2016; Kunz, Chen and Rainville, 2020) on the topic do constitute limited knowledge, especially if we compare it to the very large body of literature on the neural correlates associated with pain ratings. Except for these three studies, all the other citations pertain to behavioral studies on facial expression of pain, and do not examine the brain activity related to it. Furthermore, we believe that the complementary nature of the analyses performed in Kunz et al. (2011) and in this manuscript offers new insights into our understanding of facial expression in the context of pain. Indeed, the multivariate approach used in this study addresses some limitations present in Kunz et al. (2011) univariate analyses, mainly that it provides a quantifiable way to compare the similarity between different predictive patterns (Reddan
and
Wager,
2017). We submit that the assessment of the FEPS against several other pain-relevant signatures provides new and important information.

Furthermore, the abstract does not clearly state the aim, and the first line of the results does not match what the authors claim in the preceding line. The take-home message (last sentence) introduces the concept of a biomarker, which, as stated before, cannot be validated with the current data/experimental design. To put it in plain words, a given facial expression (or a composite score derived from a combination of expressions) cannot be a specific biomarker for pain, because a person can always mimic the same expression without feeling pain. Whether a given facial expression can be predicted from brain activity is a different issue, and whether that prediction can differentiate between painful and non-painful origins of the facial expression is another different issue. Unfortunately, neither of those issues can be tested with the current data/experimental design. The abstract would improve if the authors would circumscribe to what they actually tested, which is accurately described in the last sentence of the Introduction.

The abstract was revised accordingly. The term ‘biomarker’ was used in accordance with preceding studies in the field (see Reddan
and
Wager,
2017; Lee
et
al.,
2021). Please note that we applied the same reasoning to fluctuations in pain expression as previous studies have applied to pain ratings. Of course, we can not dismiss the possibility of someone mimicking facial expressions. Similar reasoning applies to subjective reports, as individuals can intentionally overestimate their pain experience conveyed through verbal reports. This is another case of specificity testing that cannot be addressed in the present study (see new conclusion of the abstract and discussion of limitations). The challenge of pain assessment is a classical problem within both the scientific and the clinical literature. Here, we suggest that the consideration of multiple manifestations of pain is necessary to address this challenge and will provide a more comprehensive portrait of pain-related brain function.

IntroductionI believe that the Introduction would benefit from a strict definition of what is a marker/biomarker/neuromarkers (all those terms are used in the manuscript) and what are its desirable features (validity, reliability, specificity, etc.). I also believe that the Introduction (and the rest of the text) would benefit from a critical assessment of the term "signature". The Introduction describes four existing "signatures", all of them differing in the experimental condition in which acute nociceptive pain is studied, and proposes a fifth one. Keeping with the analogy, I'm wondering whether they should be called (pain) "signatures" if there is a different one for each experimental acute pain condition, and they are so dissimilar between them when they are tested on the same condition (this dataset).

The last part of that comment raises fundamental methodological potential limitations that should be addressed in more depth in another article. That point goes beyond the scope of a research article. Regarding the stability aspect of the signatures, most of the signatures have not been studied extensively. It is thus difficult to currently assess their reliability. However, Han et al. (2022) showed high within-individual test-retest reliability for the NPS across eight different studies. Given that pain is a multidimensional experience, it is not surprising to find different patterns of activation predictive of different aspects or dimensions of the pain experience (see Čeko et al., 2022 for a similar discussion applied to negative affect).

The authors state that "As an automatic behavioral manifestation, pain facial expression might be an indicator of activity in nociceptive systems, perceptual and evaluative processes, or general negative affect." Doesn't it reflect all three of them? (and instead of or?) Why "might"?

The original sentence has been modified as follows: “As an automatic behavioral manifestation, pain facial expression is considered to be an indicator of activity in nociceptive systems, and to reflect perceptual and affective-evaluative processes” (lines 65 to 67).

MethodsThe pain scale should be described. Kunz et al. used a 0-100 scale, where 50 was the pain threshold. This is crucial to interpret the 75-80/100 score for the painful thermal intensity.

The description of the pain scale has been added to the manuscript (lines 299 to 300).

Ratings for warm and painful temperatures should be reported (ideally plotted with individual-trial/subject data). In the same line of reasoning, FACS scores should be reported as well (ideally plotted with individual-trial/subject data). It would be interesting to explore the across-trial variability of pain ratings and FACS scores. That is, do people keep giving the same ratings and making the same facial expression after 16 trials? How much variability is between trials and between subjects?

The point raised in that comment was already addressed in response to a comment made by Reviewer #1 (also see the new Figures S2 and S4; see also lines 335 to 346).

How come only painful trials are analyzed? What if the FEPS signature was the same for warm and painful stimulation, thus reflecting the settings (fMRI experiment, stimulation, etc.) rather than the brain response to the stimuli?

The point raised in that comment was already addressed in response to a comment made by Reviewer #1. There was no pain expression in the warm trials and the FEPS shows no response to warm trials. This is now illustrated in the new Figure S4B (see also lines 138 to 158).

The authors propose to predict the trial-by-trial FACS composite score from the pain ratings using a LMM. However, it is interesting that they aim for an almost constant within- and between-subject pain score (75-80/100) as stated in the Methods. This should theoretically render the linear model invalid since its first (and main) assumption would be that FACS should vary linearly with the pain score. Even if patients were not aware that the temperatures were constant across trials, the variation in pain scores should be explained by random noise for a constant stimulation intensity.

Reviewer #3 raises an important point that we need to clarify. Contrary to the expectation that FACS responses should be strongly correlated to pain ratings, we posited that these response channels depend at least in part on separate brain networks that may be differentially sensitive to a variety of modulatory mechanisms (attention, emotion, expectancy, motor priming, social context, etc.). This implies that part of the variance in FACS is independent from pain ratings. We, therefore, consider what Reviewer #3 refers to as random noise to be relevant and meaningful fluctuations reflecting endogenous processes influencing one’s experience of pain and differentially affecting various output responses.

I noticed that fMRI data was analyzed with SPM5 in the original paper (Kunz et al., 2011) and with SPM8 in this manuscript. Was fMRI data re-processed for this manuscript? Were there any differences between the original analysis and this one that might induce changes in the interpretation of results?

The data were indeed re-processed using SPM8, which was the most recent version available when we started the analyses reported here. We used trial-by-trial activation maps for MVPA, which differs from what was used in the previous study (contrast maps at the level of the conditions, not the trials). We have no reason to believe that the different versions will change the message of this manuscript since those versions do not differ significantly in terms of the fMRI preprocessing pipeline (see SPM8 release notes; https://www.fil.ion.ucl.ac.uk/spm/software/spm8/). Furthermore, the aim of this present study is not to compare the different analysis parameters implemented in SPM5 vs SPM8.

What is the rationale for including PVP in the comparison among signatures? The experimental settings in which it was devised are distant from those described here.

The inclusion of the PVP was aimed at enhancing our comparative analysis with the FEPS, as we sought to investigate the potential functional meaning of the FEPS. The PVP was developed to capture the aversive value of pain, a dimension that is conceptually proximal to the interpretation of the facial expression as a manifestation of the affective response to nociceptive pain.

The LASSO-PCR approach is, in my opinion, not a procedure for (brain) decoding in this context. It is accurately described in the section title as a method for multivariate pattern analysis, or as a variable selection and regularization method for a prediction model. Here, brain activity in specific areas related to pain processing can hardly be described as "encoded", and the method just helps select those activations relevant for explaining a certain outcome (in this case, facial expressions).

We understand the point made by reviewer #3. The term brain decoding was changed for multivariate pattern analysis in the latest version of the manuscript.

Details are missing with regards to the dataset split into training, validation, and testing.

Details about the training and testing procedure were added in the manuscript (lines 383 to 385).

This might just be ignorance from me, so I apologize in advance, but what are "contrast" fMRI images? They are mentioned three times in the text but not really described. Are they the "Pain > Warm" contrasts from the original paper?

We apologize for any confusion caused by the use of the term “contrast images” which suggests a direct comparison between two experimental conditions. We have replaced “contrast images” with “activation maps” to provide a more accurate description of the nature of the data used in the multivariate pattern analysis (lines 388 to 389).

In the "Facial expression" section, the authors run an LMM to test the association between pain ratings (response variable) and facial responses (explanatory variable). If I understand correctly, in the "Multivariate pattern analysis" section they test the association between facial composite scores (response variable) and pain ratings (explanatory variable), but they obtain different results.

The analyses were recomputed on the log transformed data, as mentioned previously in the response to reviewers 1-2. The first model (in the “Facial expression” section) used the log transformed FACS scores as a dependent variable, the pain ratings as the fixed effect, and the participants as the random effect. The results of that analysis suggested that the transformed facial expression scores were not significantly associated with the pain ratings (p = .07). The second model uses both the FEPS pattern expression scores and pain ratings as fixed effects to predict facial responses. This analysis showed the significant contribution of the FEPS to the prediction of FACS scores (p < .001) and no significant effect of the pain ratings. However, a significant interaction was found (p = .03) suggesting that the prediction of the pain facial expression by the FEPS may vary with pain ratings (i.e. moderator effect). Those results have been clarified in the “Multivariate pattern analysis” section in the Methods (lines 416 to 426).

In this same section, what are "FEPS pattern expression scores"? They are used three times in the text, but I could not find their description.

The FEPS pattern expression scores correspond to the dot product between the trial-by-trial activation maps and the unthresholded FEPS signature. This information has been added to the manuscript (lines 413 to 414).

It would not be far-fetched to hypothesize that FACS scores could be predicted using solely activity from the motor cortex. The authors attempted to do this, but only with information from M1. Why did they not use the entire motor cortex, or better, regions of the motor cortex directly linked with the AUs described in the manuscript?

The selection of the primary motor area (M1) was based on the results found in Kunz et al. (2011). In this study, M1 showed the strongest correlation with facial expression of pain. There are numerous possibilities of combinations of multiple brain regions considering a variety of criteria based on distributed networks involved in motor, affective, or pain-related processes. We limited our exploration to the region with the strongest hypothesis due to practical feasibility concerns.

Results and DiscussionAs a general recommendation, results should present individual data whenever possible. For example, the association between signatures and facial expression should be plotted using scatterplots.

We have added figures showing individual data when it was applicable (Figure S2; Figure S4).

The authors state that the LASSO-PCR model accounts for the facial responses to pain. I believe this is an overstatement, considering:- A Pearson's r of 0.49 is usually considered low/weak correlation (moderate at best). In the same line, an R2 of 0.17 means that only 17% of the variance is explained by the model.

More nuanced interpretation of the results has been added to the discussion. A section has been added to highlight the limitations of the study.

- Figure 1 needs to display individual subject data and the ideal regression line.

The model was trained using a k-fold cross-validation procedure. The regression lines thus represent the model’s prediction for each one of the 10 folds (i.e. each fold is trained and tested on a different subset of the data). A scatter plot including the ideal regression line computed across all trials and subjects was added in supplementary material to illustrate the relation between the FACS scores and the FEPS pattern expression scores (Figure S4).

- Looking at Figure 1, it is clear that the model has an intercept different from zero. This means that when the FACS score was zero (i.e., volunteers did not make any distinguishable facial expression), the model predicted a score larger than zero. This is not discussed in the manuscript, and in simple terms, it means that there are brain activation patterns when no discernible facial expression is being made by the volunteers. In the original paper by Kunz et al., two groups of subjects were categorized, and one of them was a facially low- or non-expressive group (n=13). This fact is not even mentioned in the manuscript.

The categorization in the previous report (Kunz et al., 2012) was based on a pre-experimental session. All subjects were included in the current analysis. This is now indicated in the Methods (lines 287 to 289).

- On the other end of the range in Figure 1, differences between the FACS scores near the maximum range (40) are underestimated by 23 to 33 points! I guess that the RMSE is smaller (6-7 points), because many FACS scores are concentrated on the low end of the scale.

This is a very interesting comment. A section discussing the limits of the model to predict the lower and higher FACS scores has been added in the manuscript (lines 232 to 250).

It is of course acceptable to interpret the low similarity between signatures as a sign that each signature describes a different mechanism related to pain processing. However, I believe that a complete discussion should contemplate other competing hypotheses. Considering that all signatures were developed using a similar painful thermal stimulation protocol, it is reasonable to expect larger similarities between signatures. The fact that they are so dissimilar could be a reflection of model overfit, i.e., all these signatures are just fitted to these particular experimental protocols and data, and do not generalize to brain mechanisms of pain processing.

We appreciate the pertinent observation. We have included a limitations section in which we discussed, among other considerations, the possible overfitting of models and the necessity of pursuing generalizability studies (lines 225 to 268).